# Modulating Optical Characteristics of Nanoimprinted Plasmonic Device by Re-Shaping Process of Polymer Mold

**DOI:** 10.3390/mi12111323

**Published:** 2021-10-28

**Authors:** Hirotaka Yamada, Kenji Sueyoshi, Hideaki Hisamoto, Tatsuro Endo

**Affiliations:** 1Department of Applied Chemistry, Graduate School of Engineering, Osaka Prefecture University, Sakai 599-8531, Japan; sxb02146@edu.osakafu-u.ac.jp (H.Y.); sueyoshi@chem.osakafu-u.ac.jp (K.S.); hisamoto@chem.osakafu-u.ac.jp (H.H.); 2Japan Science and Technology Agency (JST), Precursory Research for Embryonic Science and Technology (PRESTO), Tokyo 102–8666, Japan

**Keywords:** plasmonics, nanoimprint, plasmonic device, plasmonic sensor

## Abstract

Metal nanostructures exhibit specific optical characteristics owing to their localized surface plasmon resonance (LSPR) and have been studied for applications in various optical devices. The LSPR property strongly depends on the size and shape of metal nanostructures; thus, plasmonic devices must be designed and fabricated according to their uses. Nanoimprint lithography (NIL) is an effective process for repeatedly fabricating metal nanostructures with controlled sizes and shapes and require optical properties. NIL is a powerful method for mass-producible, low-cost, and large-area fabrication. However, the process lacks flexibility in adjusting the size and shape according to the desirable optical characteristics because the size and shape of metal nanostructures are determined by a single corresponding mold. Here, we conducted a re-shaping process through the air-plasma etching of a polymer’s secondary mold (two-dimensional nanopillar array made of cyclo-olefin polymer (COP)) to modulate the sizes and shapes of nanopillars; then, we controlled the spectral characteristics of the imprinted plasmonic devices. The relationship between the structural change of the mold, which was based on etching time, and the optical characteristics of the corresponding plasmonic device was evaluated through experiments and simulations. According to evaluation results, the diameter of the nanopillar was controlled from 248 to 139 nm due to the etching time and formation of a pit structure. Consequently, the spectral properties changed, and responsivity to the surrounding dielectric environment was improved. Therefore, plasmonic devices based on the re-shaped COP mold exhibited a high responsivity to a refractive index of 906 nm/RIU at a wavelength of 625 nm.

## 1. Introduction

Metal nanostructures can likely be used in various applications, such as in optical phenomenon enhancement devices [1,2,3], optical energy conversion devices [4,5], and sensors due to their unique optical properties. In particular, in sensor applications, they have been studied as fluorescence [6,7,8,9], Raman scattering enhancement elements [10,11,12,13], and as transducers that exhibit changes in optical properties from changes in the peripheral refractive index (dielectric constant) originating from biomolecules [14,15,16]. The optical uniqueness of metal nanostructures is derived from localized surface plasmon resonance (LSPR). LSPR is the resonance between the electric field of light and the collective oscillation of free electrons on the surface of metal nanoparticles [17,18,19].

Because of LSPR, when metal nanostructures are irradiated with light, a localized enhanced photoelectric field is formed, leading to light scattering at a specific wavelength. LSPR properties vary considerably depending on the material and shape of the metal nanostructures, the surrounding dielectric environment, and the existence of other oscillators [16,20,21,22].

By utilizing LSPR, the performance and function of metal nanostructures can be widely controlled. For example, as an optical sensor, the sensor sensitivity, dynamic range, and wavelength to be used can be controlled by considering the size and shape of the metal nanostructures.

Therefore, several studies have controlled the sizes and shapes of metal nanostructures, including metal nanocolloids, to regulate LSPR properties [21,23,24]. In particular, metal nanostructure arrays fabricated on flat substrates (plasmonic devices) have been studied and utilized for a wide variety of applications because of the size and shape of individual nanostructures. In addition, the inter-distance, arrangement, and orientation of the nanostructures can be controlled in detail [25,26,27,28,29].

For the fabrication of plasmonic devices, beam processes such as electron-beam lithography [30,31], extreme ultraviolet lithography [32,33], and focused ion beam lithography [34,35] are commonly used. These methods are preferable in terms of size fineness and applicability for the fabrication of nanostructures with various shapes. However, they have disadvantages in that they take a long time to fabricate large-sized nanostructure patterns (over cm^2^) and are inefficient for the repeated fabrication of the same nanostructure patterns.

Alternatively, nanoimprint lithography (NIL) has recently been used for the repeated fabrication of nanostructure patterns [36,37,38]. In NIL, because the nanostructure patterns are transcribed from patterned molds (commonly made of silicon), the shapes of the transcribed nanostructures are strongly dependent on (ideally the same as) the molds. Thus, molds with a large surface area are manufactured to repeatedly fabricate large-sized nanostructure patterns. In turn, when the sizes and shapes of the nanostructures have to be changed, different master molds have to be constructed using electron beam lithography and dry etching for each corresponding nanostructure [39,40]. Therefore, NIL, which is supposed to cut costs, requires high implementation costs when introducing new structures. Consequently, imprinted plasmonic devices lack flexibility when adjusting to nanostructures of different sizes and shapes.

To overcome these problems, in this study, the re-shaping of a secondary polymer mold by a post-etching process was performed to modulate the spectral characteristics of imprinted plasmonic devices. This process avoids the limitations of NIL in modifying its nanostructures and enables a wide range of applications from the original single mold. To evaluate the effect of the re-shaping process on the mold structure, the changes in the diameter and height of the nanopillars were measured as a function of etching time. After depositing different thicknesses of gold (Au) on the reshaped molds, the characteristics of the spectral modulation of the obtained plasmonic devices were evaluated using optical measurements. Finally, the responsivity of the plasmonic devices to the surrounding refractive index was measured to evaluate the effectiveness of the re-shaping process for the application of imprinted plasmon sensors. The effect of the size and shape changes through the re-shaping process on both the spectral characteristics and responsivities to the surrounding environment was discussed using optical simulations.

## 2. Materials and Methods

### 2.1. Materials

The COP and nanopillar mold film (FLH230/200-120) were purchased from Scivax Co., Ltd. (Kanagawa, Japan). MgF_2_ purchased from the Kojundo Chemical Lab. Co., Ltd. (Saitama, Japan) and Au purchased from Tanaka Kikinzoku Co., Ltd. (Tokyo, Japan) were used as the target materials for thermal deposition because of their relatively low melting point (approximately 650 °C and 1064 °C, respectively) and because they cause less damage to the COP mold during thermal deposition. The HNO_3_ used in the re-shaping process was purchased from FUJIFILM Wako Pure Chemical Industries, Ltd. (Osaka, Japan). The isopropanol (IPA) used in the evaluation of the responsivities of the plasmonic devices to the surrounding refractive index was purchased from Kanto Chemical Co. (Tokyo, Japan).

### 2.2. Methods

#### 2.2.1. Re-Shaping Process of Polymer Secondly, Mold

Figure 1 illustrates the reshaping process for the secondary polymer mold. Air-plasma etching was performed on a COP nanopillar film using a plasma treatment system (CUTE-1MP/R, Femto Science, Gyeoggi, Korea) for 0–20 min under the following conditions: generation power of 50 W, base pressure of 0.5 Torr, and air gas flow rate of 20 cm^3^/min. To avoid reductions in the height of the nanopillar throughout the etching process, a MgF_2_ mask with a 20 nm thickness was thermally deposited using a thermal evaporator (SVC-700TM, Sanyu Electron Co., Ltd., Tokyo, Japan) before the process began.

After plasma etching, the MgF_2_ mask was removed by dipping it into a 30% HNO_3_ aqueous solution and was diluted using ultrapure water at 70 °C for 30 min. The diameters of the etched molds were evaluated using a field-emission scanning electron microscope (FE-SEM) system (JSM-7610F, JEOL Ltd., Tokyo, Japan), and the information regarding the height direction was obtained using an atomic force microscope (AFM) system (AFM5000II, Hitachi, Ltd., Tokyo, Japan). The value changes of the diameters and heights of the nanopillars were measured at five points on the mold, and the average values were measured.

The plasmonic devices used in the experiments in the following sections were fabricated by depositing different thicknesses of Au on the re-shaped COP molds.

Throughout the etching process, the gas flow was controlled by the mass-flow controller inside the plasma treatment system, and all of the deposition processes were monitored using a quartz crystal microbalance-based deposition controller (model XTM/2 INFICON Co., Ltd., Yokohama, Japan). These apparatuses enabled the precise control of the structure to be fabricated and were expected to improve the repeatability of device fabrication.

**Figure 1 micromachines-12-01323-f001:**
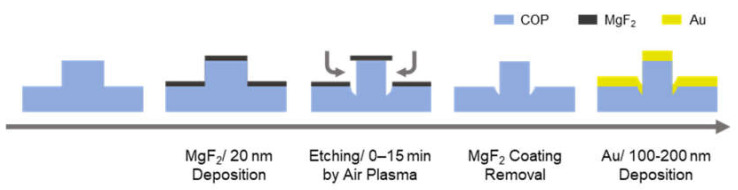
Re-shaping process of the secondary polymer mold. We thermally deposited a 20 nm MgF_2_ mask on the COP nanopillar mold and etched the nanopillar by means of air plasma gas from the side wall. After etching, we removed the MgF_2_ mask by dipping it in 30% nitric acid, and the re-shaped COP mold was obtained. Finally, to form the plasmonic device, we thermally deposited Au on the COP mold.

#### 2.2.2. Measurement of Optical Characteristics and Responsivities of Plasmonic Devices

As shown in Figure 1, the Au layer was separately deposited on the nanopillars at the tapered pointy end on the top part of the nanopillars and the pillar shaped part at the bottom of the nanopillar. During this process, there is the possibility for several LSPR modes to occur in (1) the modes where the pillar top and bottom modes are weakly coupled, (2) the modes where the pillar top and bottom modes are strongly coupled, and (3) the pillar bottom mode, which is distributed inside the COP mold. The optical characteristics of plasmonic devices were investigated by changing the Au thickness and etching time to observe the effect of the re-shaping process on the LSPR modes.

To evaluate the modulation ability of the re-shaping process on the optical characteristics of the obtained plasmonic devices, microspectroscopy was performed using an optical microscope (BX-53, Olympus Corporation, Tokyo, Japan). The broadband white light (300–1050 nm) was irradiated from a tungsten halogen light source (LS-1, Ocean Optics, Inc., Dunedin, FL, USA) through an optical fiber, and the light reflected on the surface of the plasmonic devices was introduced to a handy-type spectrometer (USB-4000, Ocean Optics, Inc., Dunedin, FL, USA) through the optical fiber. To evaluate the LSPR properties of each plasmonic device, the extinction intensity at each wavelength was calculated using the following equation:*I*_ext_ = 1−*I*_refl,plas_ / *I*_refl,flat_(1)
where *I*_ext_ is the extinction intensity, *I*_refl,plas_ is the reflection intensity from plasmonic devices, and *I*_refl,flat_ is the reflection intensity from the flat Au film.

Because light is absorbed and scattered through LSPR at a specific wavelength, LSPR modes can be found at the extinction peaks of the extinction spectra.

The responsivity of the plasmonic devices to the surrounding refractive index was also evaluated using the same microspectroscopy system. IPA aqueous solutions with different refractive indexes (concentration: 0–100% w/w, refractive index: 1.3327–1.3728) were introduced onto the plasmonic device as the control samples of the refractive index. To maintain the volume and thickness of the sample liquid during the measurement, a silicone rubber sheet with a hole of 12 mm diameter and 1 mm thickness was used as a reservoir to introduce the sample solutions, and a cover glass was placed on it to prevent evaporation. To evaluate the responsivity, the spectral change due to the change in the surrounding refractive index was measured. In the experimental setup, the quality factor of the LSPR mode was low, and the LSPR peaks were broadened. As this makes the LSPR peak unclear, the shift of inflection point was measured [41,42,43]. Here, the inflection point was calculated to be the steepest point on the right side of the LSPR peak. The responsivity was calculated from the slope of the plot between the refractive index and the inflection point shift. The responsivities of each plasmonic device were compared, and the effect of the re-shaping process was evaluated.
Responsivity [nm/RIU] = ∂*λ*_i_ / ∂*n*(2)
where λ_i_ is the wavelength of the inflection point, and n is the surrounding refractive index.

#### 2.2.3. Optical Simulation

To discuss the effect of the structural changes that occurred throughout the re-shaping process on both the spectral characteristics and responsivity to the surrounding environment, optical simulation analysis was performed. The simulation was executed using finite-difference time-domain solutions acquired from Lumerical Solutions, Inc. (Vancouver, BC, Canada).

## 3. Results

### 3.1. Re-Shaping Effect on the COP Molds

The size and shape changes of the COP molds through the re-shaping process with different etching times were evaluated using scanning electron microscopy (SEM) and atomic force microscopy (AFM) images. The SEM images shown in Figure 2 indicate a gradual reduction in the diameter of the nanopillars as the etching time increased. Furthermore, a ring-like dark tone area was observed around the nanopillars on the etched molds.

Figure 3 presents the AFM images and cross-sectional profiles. As shown, the plane surface at the top of the nanopillars remained flat even though the etching time increased. The result suggests anisotropic shape changes of nanopillars due to the MgF_2_ mask. The diameter and height of the nanopillars for each etching time (Figure 4) also indicate the successful re-shaping (reduction of diameter) of the COP mold without the loss of the nanopillar height. The diameters were measured to be 250 ± 4.3, 197 ± 5.9, 166 ± 2.8, and 130 ± 2.6 nm for etching times of 0, 5, 10, and 15 min, respectively, and the heights were measured to be 338 ± 1.4, 324 ± 6.4, 340 ± 1.5, and 327 ± 5.5 nm for etching times of 0, 5, 10, and 15 min, respectively.

In addition, from the cross-sectional profiles, the ring-like dark-toned area in the SEM images represent the pits formed at the bottom of nanopillars, as indicated by the white arrows in the figures. The pits may have formed through three steps: (1) the nanopillars are etched from the side wall, reducing the diameter; (2) the bottom area without the MgF_2_ mask is exposed with decreases in the diameter; and (3) the unmasked bottom area is etched, forming pits. From the cross-sectional profiles, the pit depths were measured to be 0, 32 ± 8.7, 105 ± 5.9, and 173 ± 8.5 nm for the etching times of 0, 5, 10, and 15 min, respectively. The nanopillar structures were not observed in the COP mold that had been etched for 20 min. This can be attributed to the fact that the pillars became excessively narrow, leading to cracks. In addition, the objects inside the pits in the SEM image likely indicate a broken part of the nanopillars (Figure 2e). From these results, the maximum time of this re-shaping process was 15 min under the present conditions for the nanopillars with heights and diameters of 338 and 139 nm, respectively.

**Figure 4 micromachines-12-01323-f004:**
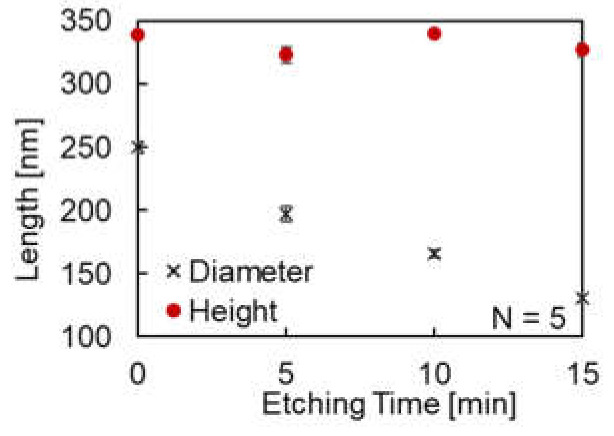
Diameter and height of nanopillars of re-shaped COP molds for each etching time. The red dots indicate height, and the black crosses indicate diameter.

### 3.2. Re-Shaping Effect on the Optical Characteristics of Plasmonic Devices Based on Processed COP Molds

Figure 5 presents the extinction spectra of plasmonic devices fabricated with different etching times and thicknesses of Au under dry conditions. When the etching time was 0 min (without re-shaping), broad extinction peaks were observed at 769, 812, and 875 nm for the Au thicknesses of 100, 150, and 200 nm, respectively. The broad and high extinction peaks are thought to be due to the overlapping of multiple peaks. Conversely, when the etching time was increased when the Au thickness was fixed, a broad peak that had split into two sharper peaks was observed. When the Au thickness was 100 nm, the peak at 769 nm was split into peaks at 657 and 811 nm, 634 and 862 nm, and 605 and 888 nm for the etching times of 5, 10, and 15 min, respectively. On the refractive index sensor application, the overlapping of extinction peaks may disturb the peak shift measurements due to the differences of the shift value of each peak. Therefore, the peak splitting behavior had a positive effect on sensing applications in dry conditions.

The peak splitting behavior (red shift of the peak at longer wavelengths and blue shift of the peak at shorter wavelengths) was not expected, especially when considering the size effect of the nanoparticles and the mode coupling efficiency. The blue shift of both peaks is likely induced by the reduction of the pillar diameter when considering the size effect. Moreover, the peak at the longer wavelength, which is expected to be assigned to the less-order mode, exhibits a larger blue shift [44,45]. Furthermore, the weakening of mode coupling and less peak splitting is expected from the formation of the pits when considering the mode coupling efficiency [46,47].

**Figure 5 micromachines-12-01323-f005:**
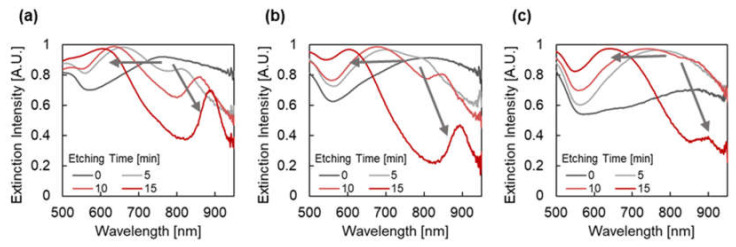
Extinction spectra of plasmonic devices fabricated by depositing Au with a thickness of (**a**) 100 nm, (**b**) 150 nm, and (**c**) 200 nm under dry conditions. The chart series in the graphs indicate the etching time (min), and the gray arrows in the graphs indicate the shift of extinction peaks.

Optical simulation analysis was performed to clarify the unexpected peak splitting behavior and to investigate the effect of pits on the LSPR characteristics of plasmonic devices. Figure 6a presents the simulated spectra of plasmonic devices with different pit depths, while the diameter was fixed at 190 nm. From the spectra, a large red shift of the longer peak wavelength was observed from 794 nm at 0 nm depth to 987 nm at 50 nm depth. The cross-sectional electric field distribution was investigated to understand the red shift of the peak. The electric field distribution indicates how the LSPR mode distributions help to understand the changes that occur in LSPR properties due to the shape change. Figure 6b,c show the cross-sectional electric field distribution at the peak wavelength of plasmonic devices with different pit depths of 0 and 50 nm and with the diameter fixed at 190 nm. From the figure, at a depth of 0 nm, the mode shifted to the longer wavelength was expected to be the pillar bottom mode distributed inside the COP mold. Furthermore, by comparing the figures, the pillar bottom mode distributed into the COP pit was observed. This indicates that the pillar bottom mode distribution shifts to the COP side when there is a high refractive index, and it derives a red shift of the peak at a longer wavelength.

According to these results, the splitting behavior observed in the experiment might be a combination of the size effect and mode coupling efficiency alone as well as the distribution of the mode to the high refractive index region [48].

**Figure 6 micromachines-12-01323-f006:**
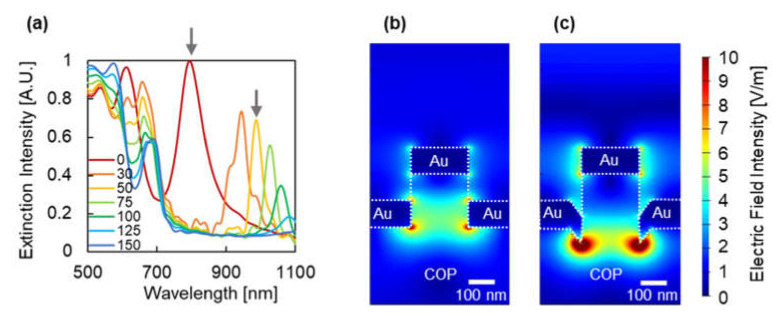
(**a**) Simulated spectra of plasmonic devices with different pit depths while the diameter is fixed at 190 nm. The chart series in the graphs indicate the pit depth (nm). Cross-sectional electric field distribution of plasmonic devices with a different pit depth of (**b**) 0 nm at the 610 nm wavelength are pointed out in Figure 6a by the left gray arrow, and (**c**) the 50 nm and at 985 nm wavelength pointed out in Figure 6a by the right gray arrow.

Figure 7 presents the extinction spectra of plasmonic devices fabricated with different etching times and thicknesses of Au in water. Compared to the extinction spectra under dry conditions, the spectra changed significantly to become complex because of the large difference in the surrounding refractive index. In contrast, the separated peak at longer wavelengths showed small responsivity to the surrounding refractive index because of the mode distribution in the COP. For example, the peak at the 895 nm wavelength for an etching time of 15 min (Figure 7a) was observed at 888 nm, and the peak shift was only 7 nm from dry to wet conditions. In addition, no peak was observed around the wavelength range of 700 to 750 nm of the spectra (gray arrows and gray dashed squares) under conditions such as an etching time of 0 min and Au thickness 100 nm; an etching time 0 and 5 min and Au thickness 150 nm; and an etching time 0, 5 and 10 min and Au thickness 200 nm.

Here, a simulation analysis was performed to identify the modes of plasmonic devices in water-like conditions (*n* = 1.335). The diameter, Au thickness, and pit depth were 230, 100, and 0 nm, respectively. Figure 8a presents the simulated spectrum, which depicts peaks in the wavelength regions of 600–670 nm, 670–750 nm, and 750–900 nm. Figure 8b–d show the cross-sectional electric field distributions assigned to each extinction peak. As shown, each extinction mode was assigned as follows: The peak in the wavelength region of 600–670 nm, where the electric field is distributed around the pillar top and small area of the pillar bottom, is assigned to the weakly coupled mode of free-electron oscillation on the pillar top (pillar top mode) and free-electron oscillation on the pillar bottom (pillar bottom mode). The peak in the wavelength regions of 670–750 nm, where the electric field is distributed between the pillar top and pillar bottom, is assigned to the strongly coupled mode of the pillar top and bottom modes. The peak in the wavelength region of 750–900 nm, where the electric field is distributed inside the pillar bottom, is assigned to the pillar bottom mode distributed inside the COP mold.

According to the results of the experiment and simulation, the experimentally obtained peak in the 700–750 nm wavelength range of the spectra, which is absent in some conditions, is likely assigned to the LSPR mode where the Au pillar and hole modes are strongly coupled. To form the LSPR mode, it was necessary to spatially separate the upper and bottom Au pillars. Therefore, when the distance between the Au pillar and Au hole is small, the Au pillar and Au hole might be connected through the deposition of the Au layer, and the peak becomes absent. As mentioned in Section 3.1, when the difference between the Au thickness and sum of the COP pillar height and pit depth is less than 254 nm; the absence of the peak was observed in this study.

**Figure 8 micromachines-12-01323-f008:**
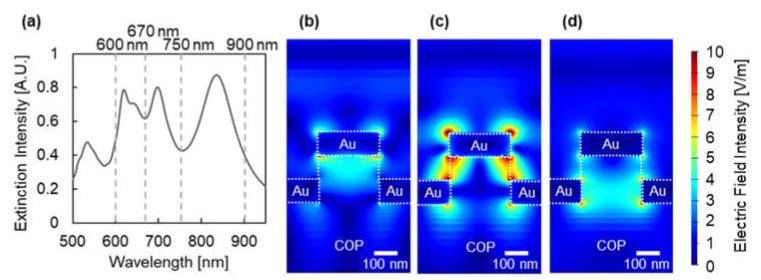
(**a**) Simulated spectra of plasmonic device with 230 nm diameter, 100 nm Au thickness, and 0 nm pit depth. The gray dashed lines at 600, 670, 750, and 900 nm wavelengths in the graph indicates the estimated divisions of extinction peaks. Cross-sectional electric field distribution of plasmonic devices at peak wavelengths of (**b**) 641 nm, (**c**) 698 nm, and (**d**) 835 nm.

### 3.3. Re-Shaping Effect on the Responctivities to the Surrounding Refractive Index of Plasmonic Devices Based on Processed COP Molds

Figure 9 shows extinction spectra of plasmonic devices varying along with the surrounding refractive index. The spectral changes were measured on all plasmonic devices with different fabrication conditions. The inflection point shifted, as indicated by gray arrows. Although the gradual shift of the inflection point could be observed in most of these spectra in the 600 to 680 nm wavelength range, spectral shift was difficult to measure in some conditions, particularly at the etching times 10 and 15 min and at a Au thickness of 100 nm and at etching time 15 min and a Au thickness 150 nm. This is because the extinction peak became unclear due to spectral overlap with the peak in the 700–750 nm wavelength range. As described in Section 3.2, the LSPR mode in the range might be the strongly coupled mode of the pillar top and pillar bottom modes. Therefore, the mode is distributed between the pillar top and pillar bottom and becomes less responsive to the surrounding refractive index due to an overlap that hides the shift in the shorter wavelength range.

Figure 10a–c illustrates the inflection point shift due to the change in refractive index under each fabrication conditions. The data of plasmonic devices with the same thickness of Au were plotted in the same graph to compare the effect of different etching times. From these figures, an increase in the shift by the surrounding refractive index due to re-shaping process was observed at the 100 and 150 nm Au thicknesses. The responsivity of the plasmonic devices was evaluated and compared using Equation (2). Because the LSPR response to the wide range of refractive indexes tends to be non-linear and quadratically even more so when plasmon modes are coupled [49,50], the responsivity was evaluated in the refractive index range of *n* = 1.3562–1.3728. Figure 10d presents a comparison of the responsivity of each plasmonic devices. The responsivity of the plasmonic devices was 482 and 698 nm/RIU for 0 and 5 min of etching time, respectively, at the 100 nm Au thickness; 513, 906, and 811 nm/RIU for 0, 5, and 10 min of etching time, respectively, at the 150 nm Au thickness; and 622, 470, 573, and 434 nm/RIU for 0, 5, 10, and 15 min of etching time, respectively, at the 200 nm Au thickness. These results indicate the improvement of responsivity by size and shape changes in the COP mold and plasmonic device when the 100 and 150 nm Au thicknesses were used.

To discuss how size and shape changes affect responsivity, the optical characteristics of the model plasmonic device resembling the fabricated structure were analyzed by simulation. Considering the results presented in Section 3.1, the diameter and pit depth were set as (A) 230 and 0 nm, corresponding to an etching time of 0 min, (B) 190 and 40 nm, corresponding to an etching time of 5 min, (C) 160 and 120 nm, corresponding to an etching time of 10 min, and (D) 140 and 190 nm, corresponding to an etching time of 15 min. In the simulation, the surrounding refractive index changed from 1.335 to 1.375. Figure 11 demonstrates the variation in the simulated extinction spectra of the plasmonic devices with the surrounding refractive index. For plasmonic devices with Au thicknesses of 150 and 200 nm, a gradual shift of the spectra was observed in the 600–680 nm wavelength range, as observed in the experiment. As mentioned in Section 3.2, when the Au thickness is 100 nm and overlaps with the LSPR mode at approximately 700 nm, the shift of the peak in the 600–680 nm wavelength range is disturbed. Therefore, to evaluate the responsivity, we used the spectral shift of the plasmonic devices with a clear peak in the 600–680 nm wavelength range (Figure 11a–d) for the 150 and 200 nm Au thicknesses. In addition, because the simulated peaks were different from the measured spectra in the experiment, the extinction peak wavelength was measured for each surrounding refractive index.

Figure 12a,b illustrates the simulated extinction peak shift due to changes in the refractive index. As shown, the responsivity was increased due to changes in the size and shape for a Au thickness of 150 nm. Here, the responsivity was estimated to be (A) 242, (B) 349, (C) 366, and (D) 300 nm/RIU. In contrast to the experimental result, the responsivity increased for a Au thickness of 200 nm. In this case, the responsivity was estimated to be (A) 312, (B) 383, (C) 379, and (D) 373 nm/RIU.

To understand how the responsivity increased and the difference between the experiment and simulation, the cross-sectional electric field distribution was simulated. Figure 13a,b illustrates the electric field distributions at a wavelength of 615 nm for (A) and a Au thickness of 150 nm, and at a wavelength of 645 nm for (C) and the same Au thickness, respectively. The change in the electric field distribution was observed in the area where the pit was formed, which was caused by the interaction between the Au at the edge of the pit and the Au at the pillar top (the area surrounded by yellow dashed lines in the figures). In these results, the formation of the pit structure contributed to the responsivity improvement. In addition, the lack of responsivity improvement for the Au thickness of 200 nm in the experiment might be explained by filling the pit with Au and by connecting the Au on the pillar top and the Au at the pillar bottom of the nanopillar.

**Figure 12 micromachines-12-01323-f012:**
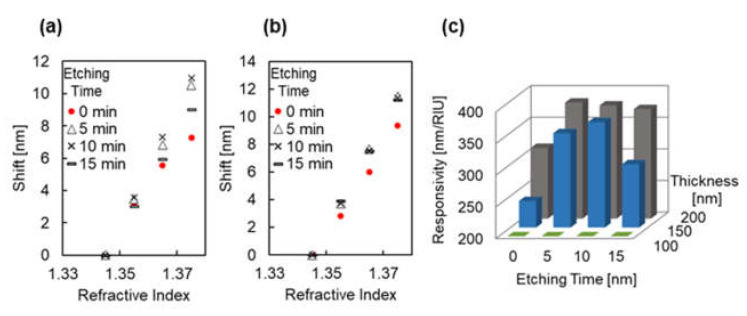
(**a**,**b**) Simulated peak shift due to refractive index changes under each condition of simulated plasmonic devices with Au layers with thicknesses of (**a**) 150 nm and (**b**) 200 nm. The red dots indicate (A), with a corresponding etching time of 0 min; the black hollow triangles indicate (B), with a corresponding etching time of 5 min; the black crosses indicate (C), with a corresponding etching time of 10 min; and the black horizonal lines indicate (D), with a corresponding etching time of 15 min. (**c**) Comparison of responsivity of each of the simulated plasmonic devices with different conditions.

**Figure 13 micromachines-12-01323-f013:**
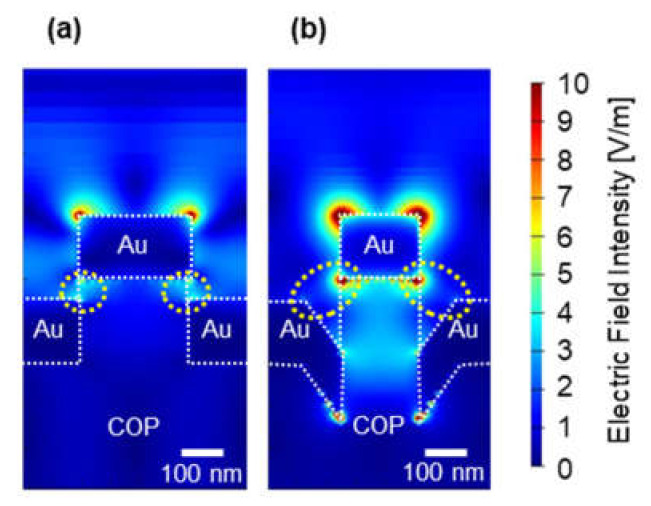
Electric field distribution (**a**) at the 615 nm wavelength for (A) and the 150 nm Au thickness and (**b**) at the 645 nm wavelength for (C) and the same Au thickness. The yellow dashed circles indicate the area where the electric field distribution changed due to pit formation.

## 4. Discussion

The re-shaping process for a secondary polymer mold was evaluated to modulate the optical characteristics of nanoimprinted plasmonic devices, with a focus on applications such as optical sensors based on changes in the surrounding dielectric environment. The size and shape changes of the nanopillar structure of the polymer mold were evaluated using SEM and AFM. The microscopy images indicated a reduction in the nanopillar diameter from 248 to 139 nm after 15 min of etching, while the height was almost constant. Furthermore, the formation of the pit at the bottom of the nanopillars was clarified as the etching progressed.

Optical measurement and simulation analysis showed that the pit structure significantly affected the spectral characteristics. The effect of the pit structure was based on the distribution of the LSPR mode to the high-refractive-index region. Therefore, changes in the size and shape by the re-shaping process modulated the optical characteristics of the plasmonic devices with three effects: size effect, LSPR mode coupling efficiency change, and mode distribution change. These effects are considered to contribute to the unexpected splitting of the peaks under dry conditions. A previous study reported the red shift of the LSPR mode by introducing a high-refractive-index material layer to the device [48]. However, in this study, the same effect occurred by changing the size and shape rather than by changing or introducing materials.

Under wet conditions, the extinction peak in the 700–750 nm wavelength range was found under certain fabrication conditions, making it difficult to measure the spectral shift based on the refractive index changes. Optical simulation revealed that the peak was assigned to the strongly coupled LSPR mode of the pillar top and pillar bottom modes, which disappears when the Au on the pillar top and Au at the pillar bottom are connected through Au deposition. The experimental result of responsivity to the surrounding refractive index revealed that the peak barely shifted and disturbed the spectral shift measurements. Hence, according to the structure of the nanoimprinted plasmonic devices studied here, a clear response can be obtained via mode vanishment by the intentional connection of the Au on the pillar top and the Au at the pillar bottom using the glancing angle and rotating deposition [51,52].

The responsivity of the plasmonic devices increased in the simulation and under some experimental conditions. From the simulated cross-sectional electric field distribution, the mode distribution change due to the pit structure variation improved the responsivity to the refractive index. The present approach to responsivity improvement depends on the 3D structure rather than on the 2D structure, and it is expected to be universally applicable to nanoimprinted plasmonic devices.

## 5. Conclusions

In this study, the optical characteristics of imprinted plasmonic devices were modulated systematically by performing the re-shaping process of the COP mold, and the relationship between size and shape changes and spectral change was clarified through simulation analysis. The plasmonic devices based on the re-shaped COP mold exhibited a high responsivity (906 nm/RIU) to the refractive index. A novel method was developed to improve the imprinted plasmonic label-free sensors by pit formation; the inflection point was used to measure the spectral shift. A high responsivity was obtained in this study, as shown in Table 1, compared to that of conventional plasmonic nanostructures. As the proposed plasmonic device operates in the visible wavelength region and allows highly productive fabrication by imprinting and application of inexpensive optical systems, it is applicable for the widespread use of highly sensitive and low-cost sensors.

## Figures and Tables

**Figure 2 micromachines-12-01323-f002:**
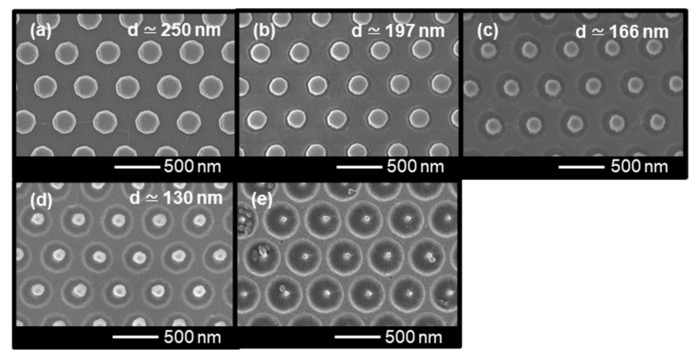
SEM images of re-shaped COP molds processed over different etching times: (**a**) 0 min, (**b**) 5 min, (**c**) 10 min, (**d**) 15 min, and (**e**) 20 min. The d value in each figure indicates the average values of measured diameters of nanopillars. The white dashed circles indicate the broken parts of nanopillars.

**Figure 3 micromachines-12-01323-f003:**
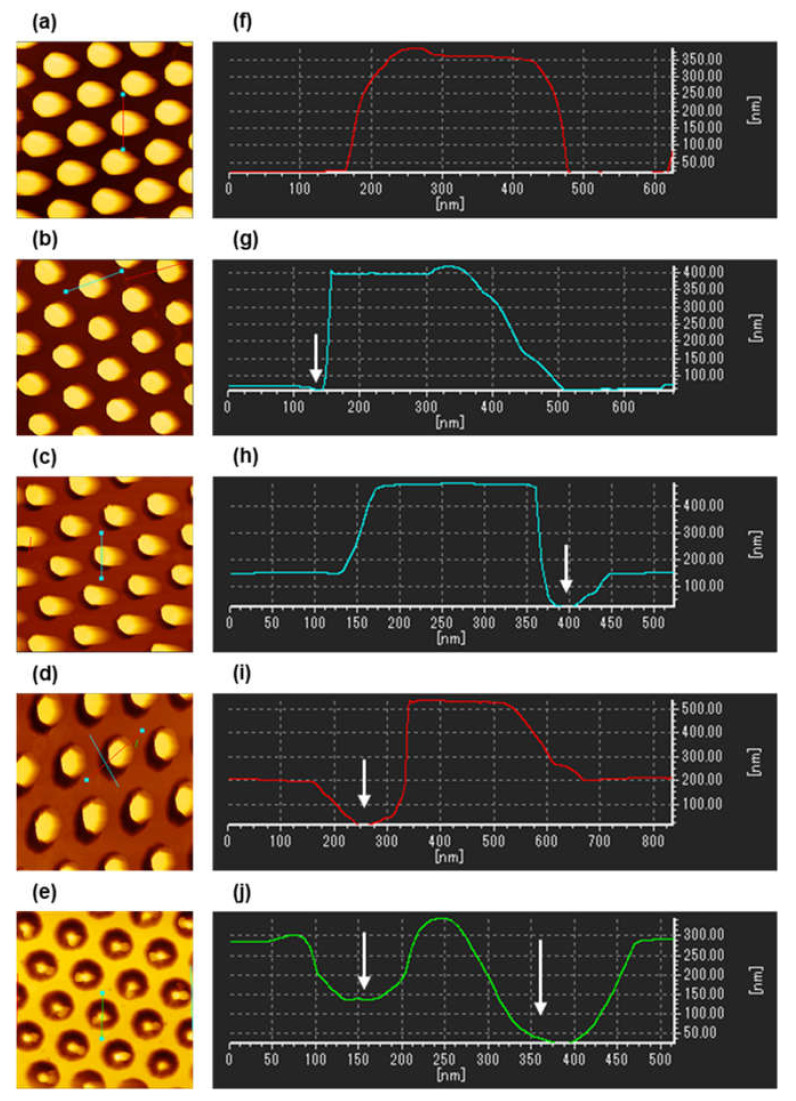
(**a**–**e**): AFM images (2 μm × 2 μm) of re-shaped COP molds processed over different etching times: (**a**) 0 min, (**b**) 5 min, (**c**) 10 min, (**d**) 15 min, and (**e**) 20 min. The line between the two light blue squares indicates the profile line required to obtain a cross-sectional profile. (**f**–**g**): Cross-sectional profile of re-shaped COP molds processed over different etching times: (**f**) 0 min, (**g**) 5 min, (**h**) 10 min, (**i**) 15 min, and (**j**) 20 min. The white arrows in the figures indicate the pits formed at the bottom of nanopillars.

**Figure 7 micromachines-12-01323-f007:**
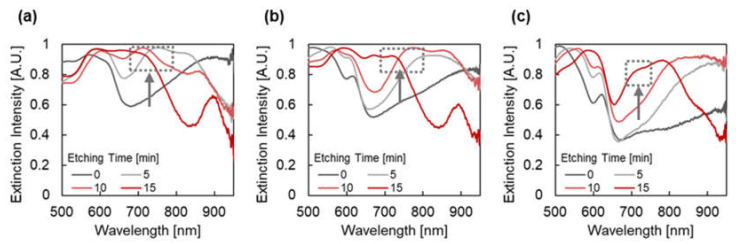
Extinction spectra of plasmonic devices fabricated through depositing Au with a thickness of (**a**) 100 nm, (**b**) 150 nm, and (**c**) 200 nm under wet conditions. The chart series in the graphs indicate the etching time (min), and the gray arrows and gray dashed squares indicate the shift of the extinction peaks, which was not found in some conditions; etching time 0 min and Au thickness 100 nm; etching time 0 and 5 min and Au thickness 150 nm; and etching time 0, 5 and 10 min and Au thickness 200 nm.

**Figure 9 micromachines-12-01323-f009:**
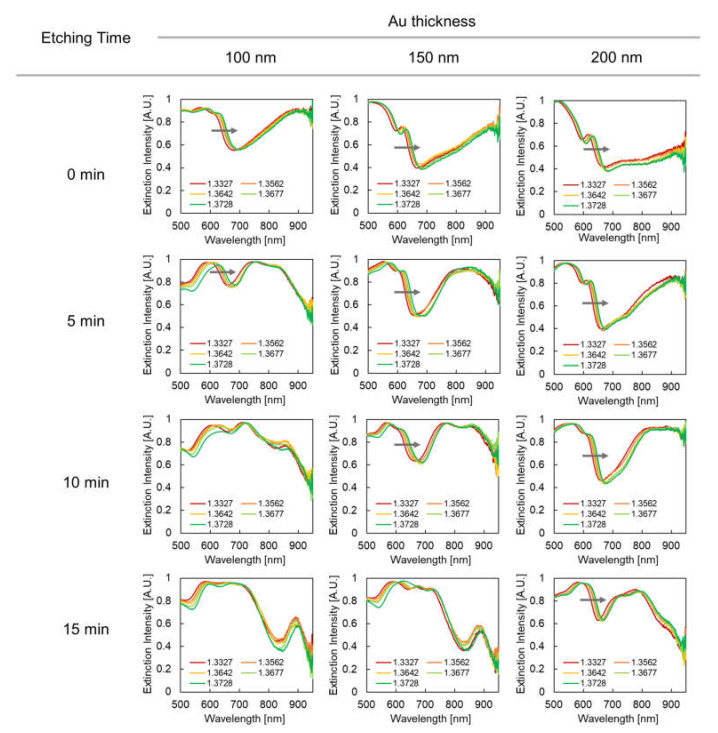
Extinction spectra of plasmonic devices based on surrounding refractive index. Columns indicate Au thickness, rows indicate etching time, and the chart series in the graphs indicates surrounding refractive index. The gray arrows indicate the spectral shift (inflection point shift) evaluated in this study.

**Figure 10 micromachines-12-01323-f010:**
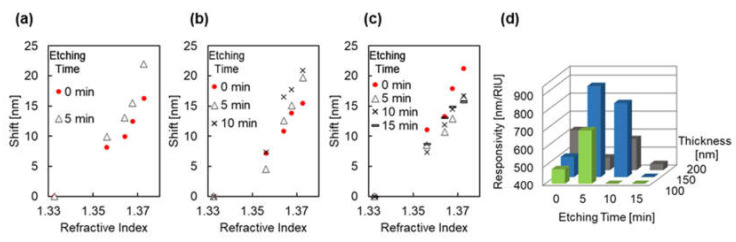
(**a**–**c**) Inflection point shifts due to refractive index changes under each fabrication condition for the plasmonic devices fabricated through depositing Au thicknesses of (**a**) 100 nm, (**b**) 150 nm, and (**c**) 200 nm. The red dots indicate an etching time of 0 min, the black hollow triangles indicate an etching time of 5 min, the black crosses indicate an etching time of 10 min, and the black horizontal lines indicate an etching time of 15 min. (**d**) The comparison of the responsivity of each plasmonic device with different fabrication conditions.

**Figure 11 micromachines-12-01323-f011:**
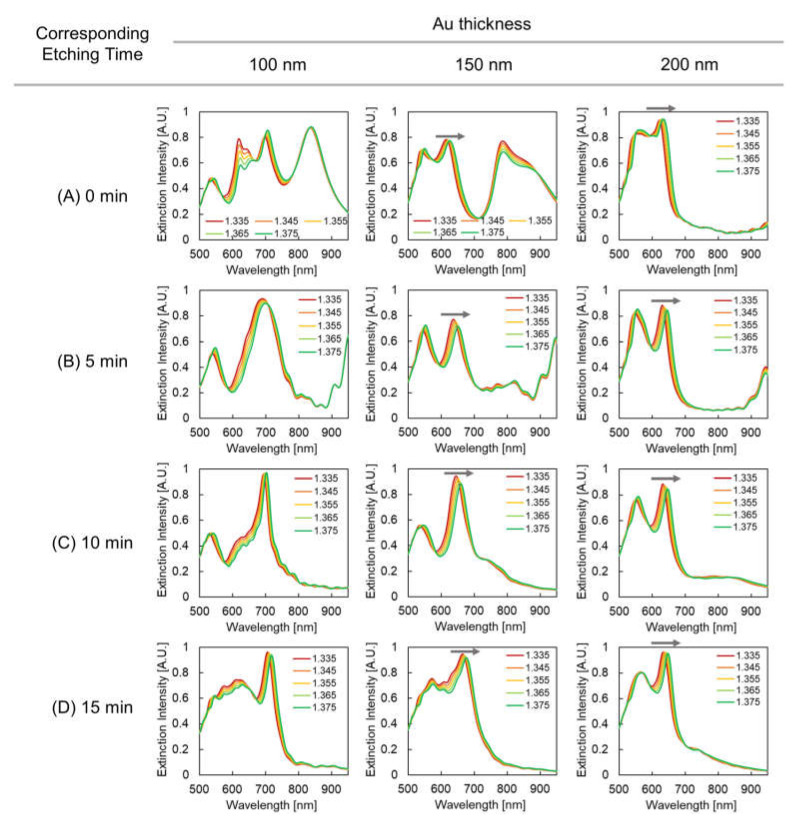
Simulated extinction spectra of plasmonic devices dependent on surrounding refractive index. Columns indicate Au thickness, and rows indicate corresponding etching times: (**A**) 0 min, (**B**) 5 min, (**C**) 10 min, and (**D**) 15 min, and the chart series in the graphs indicates the surrounding refractive index. The gray arrows in the figures indicate the spectral shift (peak shift) evaluated in this study.

**Table 1 micromachines-12-01323-t001:** Comparison of refractive index response properties of plasmonic nanostructures in previous works and this work.

Ref.	Structure	Wavelength (nm)	Responsivity (nm/RIU)
This work	Pillar-hole structure	625	906
[21]	colloidal nanorod	846	288
[53]	nanohole	710	481
[54]	nanodisk array	830	327
[55]	nanoring array	750	513
[56]	nanograting	982	886
[57]	nanocross and bar	1800	1000

## Data Availability

Data are contained within the article.

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
