# Peer review of "Modulating Optical Characteristics of Nanoimprinted Plasmonic Device by Re-Shaping Process of Polymer Mold"

_micromachines, 2021, doi:10.3390/mi12111323_

Round 1
Reviewer 1 Report
The article titled „Modulating Optical Characteristics of Nanoimprinted Plasmonic Device by Re-shaping Process of Polymer Mold” presents re-shaping process by the air etching of a polymer’s secondary mold to influence the optical characteristics of the plasmonic devices. The research is extensive, and the results are described in detail, but statistic (“in general”) is missing.
Many values do not differ strongly so without the information how many measurements and/or on how many samples the measurements were done it is difficult to not discuss with the sentence “Optical measurement and simulation analysis showed that the pit structure significantly affected the spectral characteristics” (line 390-391).
From the beginning, and most importantly, the size of the nanopillars before and after processes should be presented as an average with standard deviation (with pointed number of measured pillars). Only in such way it can be concluded that there are significant differences in diameter, high of pillars and pits depths with then influence the optical characteristics. The manuscript must be supplemented with such analysis before publishing.
Author Response
Dear Reviewer,
Thank you very much for your valuable comments and revisions for our manuscript. Our responses to your comments are described as follows.
The article titled „Modulating Optical Characteristics of Nanoimprinted Plasmonic Device by Re-shaping Process of Polymer Mold” presents re-shaping process by the air etching of a polymer’s secondary mold to influence the optical characteristics of the plasmonic devices. The research is extensive, and the results are described in detail, but statistic (“in general”) is missing.
- Thank you very much for your valuable advice. We have added the statistical data on the diameter, height of nanopillars and pits depths before and after the processes, for 5 nanopillar samples (line 110-112 shows the additional description in statical measurement). As shown in replaced Figure 4 and line 186-189 and 201-202, since the standard deviation is small compared to the size change, we concluded the valid shape change by our process.
Many values do not differ strongly so without the information how many measurements and/or on how many samples the measurements were done it is difficult to not discuss with the sentence “Optical measurement and simulation analysis showed that the pit structure significantly affected the spectral characteristics” (line 390-391).
- Thank you very much for your valuable advice. We have added descriptions in line 110-112, which describe the number of samples, and line 186-189 and 201-202, which describe the average values and standard deviations of the diameter, height of nanopillars and pits depths for each process times are indicated. In addition, we have replaced Figure 4 so that the average values with standard deviations of 5 nanopillar samples for each process times are indicated.
- From these data on the average values and standard deviations, the change on the average of diameter of nanopillars and pits depths by our process showed significant change (over 3 times of standard deviation), and that influenced the optical characteristics.
From the beginning, and most importantly, the size of the nanopillars before and after processes should be presented as an average with standard deviation (with pointed number of measured pillars). Only in such way it can be concluded that there are significant differences in diameter, high of pillars and pits depths with then influence the optical characteristics. The manuscript must be supplemented with such analysis before publishing.
- We have added the average values and standard deviations of the diameter, height of pillars and pits depths for each process times in line 186-189 and 201-202, and replaced Figure 4. Since the systematic size changes on the diameters and pit depths are over 3 times of standard deviation, we concluded the changes are significant enough to modulate optical characteristics.
Based on your valuable advice, we have revised our manuscript.

Reviewer 2 Report
This manuscript focuses on the use of chemical etching of a nanopatterened polymer substrate, produced via lithography, to induce variations in the plasmonic properties of the material upon subsequent coating with gold. The paper is a welcome addition to the literature as, for example, there is still a need to develop SERS-supporting plasmonic substrates. In this regard, the authors should consider enhancing the manuscript prior to final publication by discussing the reproducibility of the etching process and resulting plasmonic properties.
- Figure 4: Please include error bars on the determination of pillar diameter and height. How reproducible is this process? Were the measurements acquired on multiple substrates across different preparations? How many pillars were measured for each sample?
- In general, a discussion of the repeatability of both the fabrication process and the resulting plasmonic properties of the re-shaped COP materials would be of interest to the readers.
- Section 3.2: Some additional discussion to better explain the shapes of the extinction curves would be useful to the reader. For example, the appearance of more well-defined bands in Figure 5a is discussed. At shorter or no etch times, is the broad, high extinction due to overlapping LSPR modes? It may also be useful to mention why the ability to tune these properties is important for eventual use of the material. Are there situations in which one would prefer the 0 min etch profile over the 15 min etch and vice-versa?
Author Response
Dear Reviewer,
Thank you very much for your valuable comments and revisions for our manuscript. Our responses to your comments are described as follows.
This manuscript focuses on the use of chemical etching of a nanopatterened polymer substrate, produced via lithography, to induce variations in the plasmonic properties of the material upon subsequent coating with gold. The paper is a welcome addition to the literature as, for example, there is still a need to develop SERS-supporting plasmonic substrates. In this regard, the authors should consider enhancing the manuscript prior to final publication by discussing the reproducibility of the etching process and resulting plasmonic properties.
- Figure 4: Please include error bars on the determination of pillar diameter and height. How reproducible is this process? Were the measurements acquired on multiple substrates across different preparations? How many pillars were measured for each sample?
- Thank you very much for your valuable advice. We have replaced Figure 4 so that the average values with standard deviations. We have measured 5 nanopillars from each mold sample. And all etching processes were performed on the mold of 3 cm×3 cm area (relatively large area), with the single preparation condition.
- On the reproducibility of etching process, since the etching rate and depth are determined by target material and etching condition, as long as mold material is the same and the condition of etching gas plasma generation is constant, we defined the process reproducible.
- In general, a discussion of the repeatability of both the fabrication process and the resulting plasmonic properties of the re-shaped COP materials would be of interest to the readers.
- Thank you very much for your valuable advice. Certainly, the discussion in repeatability is necessary. We take the repeatability can be supported by precisely controlling the structure to be fabricated by the apparatus.
- Through the etching process, the etching condition was controlled by the plasma treatment system, including mass-flow controller inside it. In addition, the deposition processes of all materials were monitored using a quartz-crystal microbalance-based deposition monitor (model XTM/2, INFICON , Ltd., Yokohama, Japan). We have described it in line 115-119 in our manuscript.
- Section 3.2: Some additional discussion to better explain the shapes of the extinction curves would be useful to the reader. For example, the appearance of more well-defined bands in Figure 5a is discussed. At shorter or no etch times, is the broad, high extinction due to overlapping LSPR modes? It may also be useful to mention why the ability to tune these properties is important for eventual use of the material. Are there situations in which one would prefer the 0 min etch profile over the 15 min etch and vice-versa?
- Thank you very much for your valuable comment. We have added the description in line 217-218 and 222-225 in our manuscript, describing the behavior of extinction spectra. Certainly, the broad extinction peak at less etching time is due to the overlapping of LSPR peaks.
- As we additionally described in line 222-225, since the overlapping of the peaks disturbs to measure shift of each peak, the tuning of extinction spectra to make the modes individual has positive effect in sensing application at dry condition.
- The situations in which the less etching time is preferred may be the fluorescence enhancement and optical energy transfer at the overlapped wavelength is required. However, the fluorescence enhancement factor and energy transfer efficiency cannot be discussed only from the extinction spectrum shape, it may not be meaningful in this manuscript.
Based on your valuable advice, we have revised our manuscript.

Round 2
Reviewer 1 Report
The authors have been improved manuscript to such form that can be published.
Just please double check in the line 201 the first value if it is correct (0,32+/- 8.7).